# Host immunity and the colon microbiota of mice infected with *Citrobacter rodentium* are beneficially modulated by lipid-soluble extract from late-cutting alfalfa in the early stages of infection

**K. Fries-Craft**[1], **J. M. Anast**[1,2], **S. Schmitz-Esser**[1,2], **E. A. Bobeck**[1]*

**1** Department of Animal Science, Iowa State University, Ames, Iowa, United States of America,
**2** Interdepartmental Microbiology Graduate Program, Iowa State University, Ames, Iowa, United States of America

* eabobeck@iastate.edu

**Data Availability Statement:** 16S rRNA sequences are available under the BioProject ID

## Abstract

Alfalfa is a forage legume commonly associated with ruminant livestock production that may be a potential source of health-promoting phytochemicals. Anecdotal evidence from producers suggests that later cuttings of alfalfa may be more beneficial to non-ruminants; however, published literature varies greatly in measured outcomes, supplement form, and cutting. The objective of this study was to measure body weight, average daily feed intake, host immunity, and the colon microbiota composition in mice fed hay, aqueous, and chloroform extracts of early (1st) and late (5th) cutting alfalfa before and after challenge with *Citrobacter rodentium*. Prior to inoculation, alfalfa supplementation did not have a significant impact on body weight or feed intake, but 5th cutting alfalfa was shown to improve body weight at 5- and 6-days post-infection compared to 1st cutting alfalfa (*P* = 0.02 and 0.01). Combined with the observation that both chloroform extracts improved mouse body weight compared to control diets in later stages of *C. rodentium* infection led to detailed analyses of the immune system and colon microbiota in mice fed 1st and 5th cutting chloroform extracts. Immediately following inoculation, 5th cutting chloroform extracts significantly reduced the relative abundance of *C. rodentium* (*P* = 0.02) and did not display the early lymphocyte recruitment observed in 1st cutting extract. In later timepoints, both chloroform extracts maintained lower splenic B-cell and macrophage populations while increasing the relative abundance of potentially beneficially genera such as *Turicibacter* (*P* = 0.02). At 21dpi, only 5th cutting chloroform extracts increased the relative abundance of beneficial *Akkermansia* compared to the control diet (*P* = 0.02). These results suggest that lipid soluble compounds enriched in late-cutting alfalfa modulate pathogen colonization and early immune responses to *Citrobacter rodentium*, contributing to protective effects on body weight.

PRJNA598236. All other relevant data are within the paper and its Supporting Information files.

**Funding:** The authors received no specific funding for this work.

**Competing interests:** The authors have declared that no competing interests exist.

## Introduction

Phytochemicals are a class of natural compounds that may improve livestock health through direct and indirect action on the host immune system and intestinal microbiota [1, 2]. One dietary phytochemical source is alfalfa, a documented source of bioactive compounds associated with a number of health benefits such as saponins [3, 4], phytoestrogens/flavonoids [5, 6], and non-cellulosic polysaccharides [7, 8]. Alfalfa, a perennial legume forage, allows for multiple harvests to occur within a growing season. While an established ruminant livestock feed ingredient, alfalfa is used less often in non-ruminant diets. Dietary alfalfa inclusion for non-ruminant livestock is limited due to the high indigestible fiber content, reducing dietary energy density. While the fiber content has variable animal performance impacts, it may provide a substrate for microbial fermentation to modulate poultry and swine intestinal microbiota in favor of beneficial community members [9–14]. Anecdotal evidence from swine producers suggests that feeding late-cutting alfalfa may confer greater health benefits to sows and piglets compared to earlier cuttings. The use of alfalfa extracts can possibly circumvent fiber-associated limitations while still conferring immunological benefits. In poultry and swine, crude aqueous alfalfa extract and protein-rich concentrate increased lymphocyte proliferation while chloroform extracts improved mouse survival during LPS challenge and suppressed production of pro-inflammatory interleukin (IL)-6 *in vitro* [15–18].

Compounds that may be enriched in these extracts have documented impacts on the immune system and intestinal microbiota. Alfalfa-derived non-cellulosic polysaccharides isolated from alfalfa reduced pro-inflammatory cytokine expression in cultured murine macrophages, while soybean isoflavones and saponins from other plant sources have been shown to increase peripheral blood lymphocytes, proportions of cytotoxic T-cells, serum immunoglobulin (Ig)G, and intestinal IgA-secreting cells in swine, poultry, and mice [7, 8, 19–23]. In rodent models of intestinal and metabolic disease, saponins, flavonoids, and polysaccharides from various sources altered the relative abundance of microbial phyla to profiles more similar to healthy controls [24–29]. Factors such as plant maturity, season, cutting, and time of harvest have documented impacts on alfalfa's nutritional and phytochemical profiles, suggesting that feeding different cuttings may differentially impact immunity and the microbiota [6, 30–34].

While alfalfa supplementation has documented benefits, the use of livestock models is limited by the lack of reagents available to descriptively assess immunological and microbiological outcomes. In this study, mice were used to evaluate changes to the immune system and colon microbiota due to their well-characterized microbiome, wide variety of available immunological reagents, and the ability to select dietary treatments of interest for future livestock work [35, 36]. To assess alfalfa's effect in both healthy and pathogen-challenged animals, rodent-specific *Citrobacter rodentium* was administered because it has well-documented intestinal microbiota impacts, a known infection timeline, and colonizes the rodent digestive tract more successfuly than *Salmonella* or *Escherichia coli* [37–39]. The objective of the work presented here was to assess body weight (BW) and average daily feed intake (ADFI) as general health responses in mice fed early (1st) or late (5th) cutting alfalfa as hay, aqueous extract (water-soluble), and chloroform extract (lipid-soluble) before and after *C. rodentium* challenge. Noted trends and responses in BW and ADFI resulted in detailed examination of chloroform extracts and their impacts on immunity and the colon microbiota before and after pathogen challenge.

## Materials and methods

All animal protocols were approved by the Iowa State University Institutional Animal Care and Use Committee (Protocol # 11-17-8643-M). All euthanasia procedures used isofluorane gas anesthesia with cervical dislocation as a secondary method. Mice were monitored daily

during the infection period with the loss of the righting reflex defined as a humane endpoint requiring euthanasia before a planned sampling timepoint. No animals displayed behaviors related to this humane endpoint and none were found dead during the 35d study.

## Experimental design

A total of 163 6-week-old female C57BL/6J mice were obtained from Jackson Laboratories (Bar Harbor, ME) and housed in 45 Innovive cages (2–4 mice/ cage; Innovive Inc., San Diego, CA) using an ear punch to identify individuals within each cage. Following their arrival at Iowa State University, animals were given a 7d acclimation period to allow stabilization of the microbiota after transport and diet change to the Teklad Global 18% Protein Rodent Diet (Envigo Teklad, Madison, WI). After the adaptation period (d0), spleens and colon digesta were collected from 9 mice to establish a baseline and the remaining mice were randomly assigned to 1 of 7 dietary treatments (22 mice/ treatment). These treatments consisted of the Teklad Global 18% Protein Rodent Diet without alfalfa (control) or supplemented with 9% hay, 0.25% aqueous extract, or 0.25% chloroform extract of 1st or 5th cutting alfalfa (Table 1). All diets were formulated to be isocaloric and isonitrogenous by Envigo Teklad and offered to mice *ad libitum* in a pelleted form.

The study was divided into a 14d feed enrichment period and 21d infection period with *C. rodentium*, totaling 35d overall. Mice weighing 18-20g are most susceptible to *C. rodentium* and female 6-week-old mice were used to ensure that mice would be in this BW range after the 2-week feed adaptation period [38]. Following the feed adaptation period, 6 mice/ treatment were euthanized for tissue sampling and remaining animals were inoculated by oral gavage with 200μl of $2 \times 10^{10}$ colony forming units (CFU) of *C. rodentium*. Preparation of the inoculum was done according to methods published by Crepin *et al*. [38]. Briefly, *C. rodentium* strain DBS100 (ATCC 51459; Manassas, VA) was cultured overnight in 15ml lysogeny broth (LB) at 37˚C in a shaking incubator (200 rpm). The next morning, cultures were centrifuged (3000 x g for 10 minutes) and bacteria were resuspended in 15ml sterile phosphate-buffered saline (PBS). 100μl of resuspended bacteria was serially diluted and plated on LB to determine CFUs. Following inoculation, 4 mice/ treatment were euthanized for tissue sampling on 4, 8, 14, and 21d post-infection (dpi) in accordance with timelines published by Crepin *et al*. [38]. The trial concluded at 21dpi when the last subset of mice was euthanized.

**Table 1. Nutrient composition of experimental diets fed to female C57BL/6 mice for the 14d feed enrichment period and 21d infection period.**

| Diet | Reference Number[1] | Protein (%) | Carbohydrate (%) | Fat (%) | Kcal/g | % kcal from Protein | % kcal from Carbohydrate | % kcal From Fat |
|---|---|---|---|---|---|---|---|---|
| Control | TD.00588 | 18.2 | 48.0 | 5.8 | 3.2 | 22.9 | 60.5 | 16.6 |
| 1st Cutting Hay | TD.170994 | 18.2 | 46.4 | 5.5 | 3.1 | 23.7 | 60.3 | 16.0 |
| 5th Cutting Hay | TD.170995 | 18.7 | 47.3 | 5.5 | 3.1 | 23.9 | 60.4 | 15.7 |
| 1st Cutting Aqueous Extract | TD.170996 | 18.1 | 47.9 | 5.8 | 3.2 | 22.9 | 60.5 | 16.6 |
| 5th Cutting Aqueous Extract | TD.170997 | 18.1 | 47.9 | 5.8 | 3.2 | 22.9 | 60.5 | 16.6 |
| 1st Cutting Chloroform Extract | TD.170998 | 18.1 | 47.9 | 5.8 | 3.2 | 22.9 | 60.5 | 16.6 |
| 5th Cutting Chloroform Extract | TD.170999 | 18.1 | 47.9 | 5.8 | 3.2 | 22.9 | 60.5 | 16.6 |

[1] All diets were formulated and prepared in a pelleted form by Envigo Teklad (Madison, WI).

## Body weight and feed intake monitoring

Individual mouse BW was recorded on arrival, d0, d14, and daily following *C. rodentium* inoculation. Feeders were replenished when necessary and feed intake for each cage was calculated by subtracting leftover feed from both the feeder and cage bottom from the amount of added feed. The total FI from d0-14, 0-4dpi, 4-8dpi, 8-14dpi, and 14-21dpi was calculated and presented as ADFI per mouse. Following *C. rodentium* inoculation, the return to pre-inoculation BW was used as a general indicator of recovery and used to identify specific treatments for downstream analysis of the immune system and microbiota.

## Flow cytometry

Mouse spleens were gently homogenized in PBS and filtered through a sterile 70μm strainer. Red blood cells were lysed using ACK lysing buffer (Gibco, Fisher Scientific, Hampton, NH) and the obtained cells were washed twice in PBS. Splenocytes were resuspended in RPMI, enumerated using a hemocytometer and stored at -80˚C in cryotubes with heat-inactivated bovine calf serum (BCS) supplemented with 7.5% DMSO until analysis.

To determine splenic immune cell profiles, splenocytes were stained for extracellular markers and intracellular cytokines. Prior to multi-color flow cytometric analysis, cells were thawed, counted, and cultured overnight in RPMI (Fisher Scientific) with 10% BCS and 1X penicillin/ streptomycin at 37˚C, 5% $CO_2$, and 90% humidity. Cells were plated at a density of $10 \times 10^6$ cells/ well in 24-well culture plates. After overnight culture, cells were enumerated and cytokine production was stimulated for 4 hours at the previously described culture conditions using the BioLegend Cell Activation Kit with Brefeldin A prepared according to manufacturer's instructions (Cat. No. 423304). Following stimulation, cells were collected, aliquoted into flow cytometry tubes, and blocked for 10 minutes at 4˚C using mouse FC block (Cat. No. 553142, BD, San Jose, CA) diluted according to manufacturer's instructions. Cells were then washed in PBS and stained for extracellular markers diluted in cell staining buffer (Cat. No. 420201, BioLegend). The extracellular markers used were: B220 Alexa Fluor® 488 (clone RA3-6B2; rat IgG2a,κ), F4/80 PerCP-Cy5.5 (clone BM8; rat IgG2a,κ), CD11b PE/Cy7 (clone M1/70; rat IgG2b,κ), CD4 Alexa Fluor® 700 (clone GK1.5; rat IgG2b,κ), CD3 Pacific Blue (clone 17A2; rat IgG2b,κ), Ly-6G Brilliant Violet™ 510 (clone RB6-8C5; rat IgG2b,κ), and CD8α Brilliant Violet® 785 (clone 53–6.7; rat IgG1a,κ; BioLegend).

Fluorescence minus one (FMO) staining protocols were used with appropriate isotype controls to account for non-specific binding. Following incubation at 4˚C for 30 minutes in the dark, cells were washed in PBS, fixed, and permeabilized according to the manufacturer's instructions using the eBioscience™ Foxp3/ transcription factor staining buffer kit (Cat. No. 00-5523-00; Thermo-Fisher Scientific Corporation, Carlsbad, CA). Cells were stained for intracellular cytokines diluted in permeabilization buffer for 30 minutes at room temperature in the dark. Antibodies for intracellular cytokines were interferon (IFN)-γ APC (clone XMG1.2; rat IgG1,κ), tumor necrosis factor (TNF)-α PE (clone MP6-XT22; rat IgG1,κ), IL-17A Brilliant Violet™ 650 (clone TC11-18H10.1; rat IgG1,κ), and IL-22 Alexa Fluor® 647 (clone Poly5164; goat polyclonal IgG; BioLegend). After staining, cells were washed twice in permeabilization buffer and resuspended in cell staining buffer. Cells were stored in the dark at 4˚C until immune cell populations could be analyzed using a BD FACSCanto™ cytometer (BD Biosciences). Analysis of cell population data obtained from the cytometer was completed using FlowJo 10.5.0 software.

## DNA extraction and sequencing

Colon digesta was collected by incising the tissue and transferring the contents into sterile tubes using sterile forceps before freezing at -80˚C. DNA was extracted from the colon digesta

using the DNeasy PowerLyzer PowerSoil Kit (Qiagen, Hilden, Germany) according to the manufacturer's instructions. A Nanodrop 2000 spectrophotometer (Thermo Fisher Scientific, Waltham, MA) was used to quantify the extracted DNA before being frozen at -20°C. Prior to PCR amplification and sequencing, extracted DNA was diluted to approximately 30ng/μl. Microbiota sequencing was conducted using a protocol designed to amplify bacteria and archaea in the DNA facility at Iowa State University (Ames, IA) [40]. Briefly, genomic DNA from each sample was amplified using Platinum™ Taq DNA Polymerase (Thermo Fisher Scientific, Waltham, MA) with one replicate per sample using universal 16S rRNA gene bacterial primers [515F (5′-GTGYCAGCMGCCGCGGTAA-3′ [41], and 806R (5′-GGACTACNVGGGTW TCTAAT-3′ [42]] for the variable region V4, as previously described [43]. All samples underwent PCR with an initial denaturation step at 94°C for 3 min, followed by 45s of denaturing at 94°C, 20s of annealing at 50°C, and 90s of extension at 72°C. This was repeated for 35 cycles and finished with a 10 min extension at 72°C. PCR products were purified with the QIAquick 96 PCR Purification Kit (Qiagen Sciences Inc, Germantown, MD) according to the manufacturer's recommendations. PCR bar-coded amplicons were mixed at equal molar ratios and used for Illumina MiSeq paired-end sequencing with 150 bp read length and cluster generation with 10% PhiX control DNA on an Illumina MiSeq platform (Illumina Inc., San Diego, CA).

## Statistics

Mouse BW and ADFI data were analyzed using the following statistical model:

$$y_{(i)jkl} = \mu + Con_i + Cut_{(i)j} + F_{(i)k} + (Cut \times F)_{(i)jk} + d0BW_{(i)jkl} + e_{(i)jkl}$$

Where y is the dependent variable (BW or ADFI), μ is the overall mean, $Con_i$ is the effect of the control at the [i]th level (i = 1), $Cut_{(i)j}$ is the fixed effect of the [j]th level of alfalfa cutting nested within the control (1[st] or 5[th]; j = 2), $F_{(i)k}$ is the fixed effect of the [k]th level of form nested within the control (hay, aqueous extract, or chloroform extract; k = 3), $(Cut \times F)_{(i)jk}$ is the fixed effect of the interaction between cutting at the [j]th level and form at the [k]th level nested within the control, $d0BW_{(i)jkl}$ is the covariate of d0 BW associated with each observation, and $e_{(i)jkl}$ is the random error. This model was used due to the $2 \times 3 + 1$ factorial treatment design with the control, 2 cuttings of alfalfa, and 3 supplementation forms.

Data obtained by flow cytometry were analyzed using the following statistical model:

$$y_{ij} = \mu + \tau_i + e_{ij}$$

In this model, y is the dependent variable (cell population), μ is the overall mean, $\tau_i$ is the effect of treatment at the [i]th level (control, 1[st] cutting chloroform extract, or 5[th] cutting chloroform extract), and $e_{ij}$ is the random error associated with the [j]th replicate (mouse) from the [i]th level of treatment.

Both models were analyzed using the MIXED procedure of SAS 9.4 (SAS Institute, Cary, NC). The Satterthwaite method for degrees of freedom and the repeated statement by treatment group were used to analyze data under the assumption of unequal variance between treatments. Significance was denoted at $P \leq 0.05$.

## Microbiota sequencing data analysis

Sequencing data were assessed for quality and screened using mothur (v.1.40.4) [44]. Prior to clustering into operational taxonomic units (OTUs), paired-end reads were merged and sequences with ambiguous bases were removed. Sequences shorter than 250bp and longer than 255bp were removed, in addition to those with > 8 identical consecutive bases. The

remaining sequences were then randomly subsampled to 20,000 sequences/ sample. Unique sequences meeting these criteria were aligned to the SILVA v132 database. Potential chimeric sequences were removed using Vsearch and sequences within 2 mismatches of the aligned sequences were also removed before being clustered into OTUs at 97% similarity, resulting in a total of 484,558 OTUs. The SILVA SSU reference database version 132 was used as taxonomic reference [45]. Whole community comparisons were done using analysis of similarities (ANOSIM), while differences between treatments at an OTU level were analyzed using linear discriminant analysis (LDA) effect size (LEfSe) [46] implemented in mothur.

## Results

### BW and ADFI

The study arrangement presented here allowed for the assessment of baseline responses to alfalfa supplementation in addition to determining responses during infection with a rodent-specific pathogen. No BW differences were observed during the feed enrichment period (Fig 1). Similarly, no differences in ADFI were observed during the enrichment or later timepoints during *C. rodentium* challenge (Fig 2); however, significant reductions in ADFI for mice fed aqueous extract versus hay from 0-4dpi and 4-8dpi were noted (Fig 2A). Regardless of cutting, mice fed hay-supplemented diets ate 0.5g ± 0.1g (18.1%) more than mice given diets with aqueous extract in the first 4dpi ($P = 0.005$; Fig 2A). From 4-8dpi, feeding hay increased mouse ADFI by 0.5 ± 0.2g (17.2%) compared to mice fed aqueous extracts ($P = 0.04$). In the early timepoints following inoculation (0-4dpi and 4-8dpi), BW and ADFI were expected to

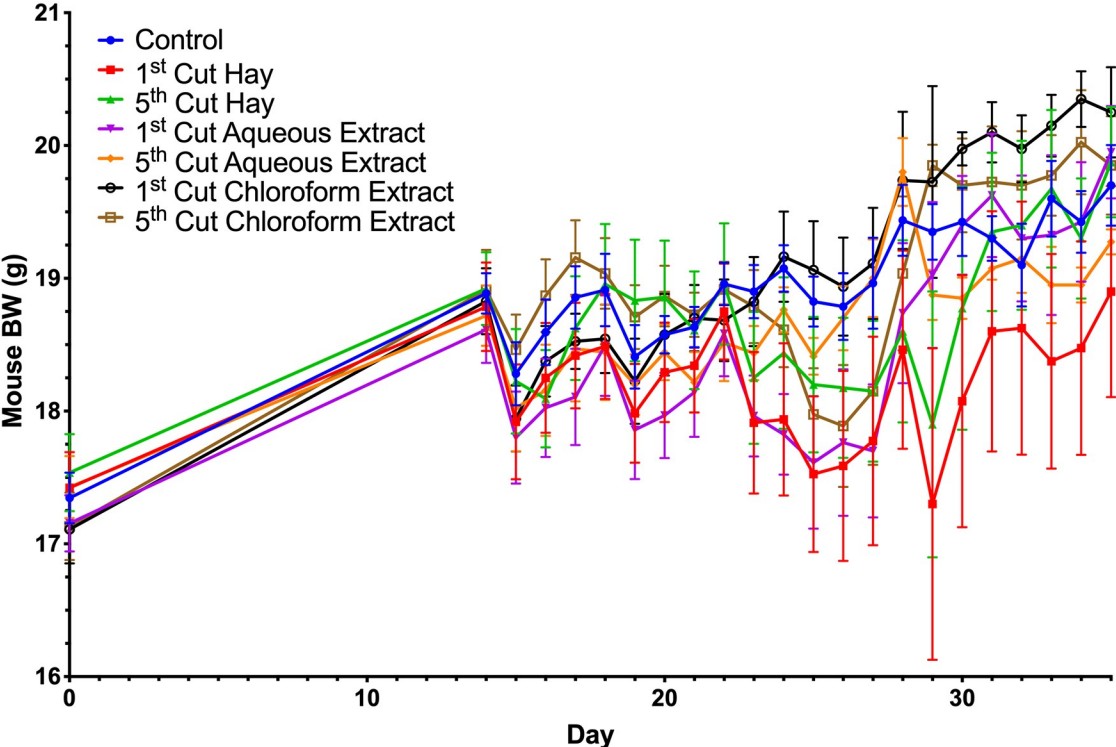

**Fig 1. Body weights of mice fed different forms of 1st and 5th cutting alfalfa before and after *Citrobacter rodentium* inoculation.** Data represent mean BW ± SEM. Key timepoints for blood and tissue sampling are marked by black arrows.

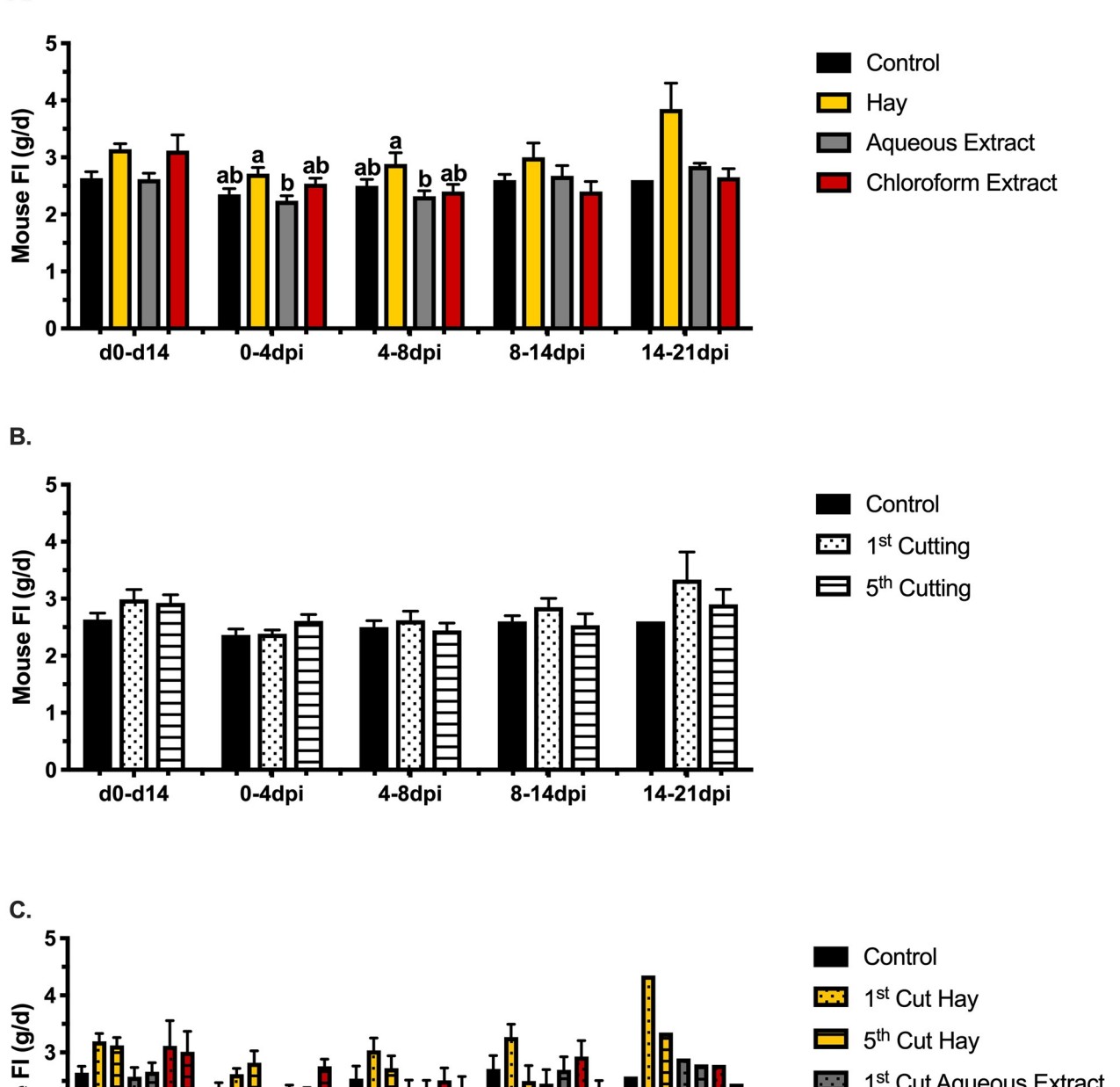

**Fig 2. Average daily feed intake of mice fed different forms of 1st and 5th cutting alfalfa.** Effects of alfalfa supplementation are separated into (**A**) form, (**B**) cutting, and (**C**) the interaction of form × cutting. Data are represented as the mean ADFI ± SEM. Bars with different letter superscripts are significant at $P \leq 0.05$.

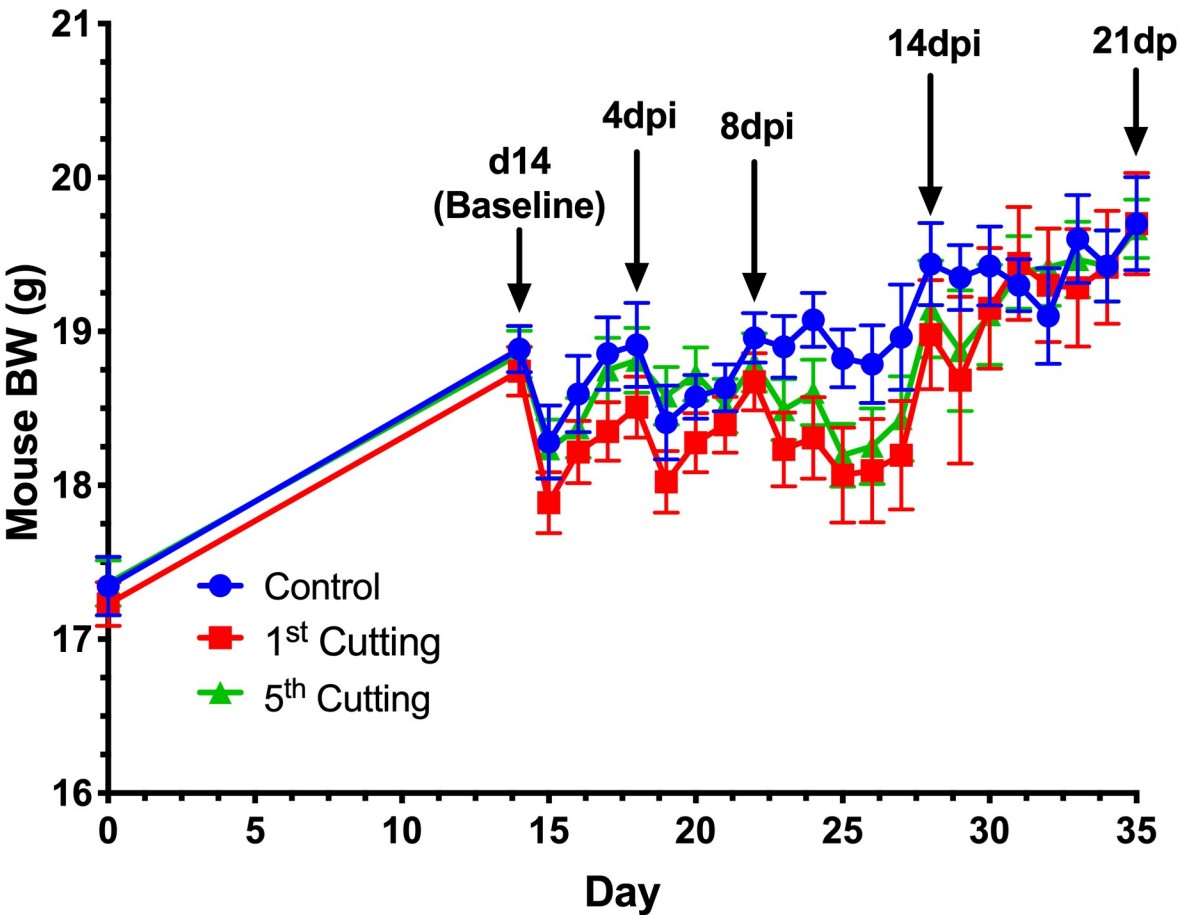

**Fig 3. The main effect of alfalfa cutting on mouse BW in comparison to the unsupplemented control.** Data represent mean BW ± SEM. Key timepoints for blood and tissue sampling are marked by black arrows.

drop with recovery occurring as *C. rodentium* was cleared during later timepoints (8-14dpi and 14-21dpi).

Post-inoculation, ADFI was unchanged by alfalfa-supplemented diets compared to the control; however, varying significant responses and trends in BW were observed (Figs 2–4). In the first 4dpi, feeding 5th cutting alfalfa positively impacted BW, regardless of supplementation form (Fig 3) and significant differences between the two cuttings were observed within the first 7dpi. Compared to 1st cutting alfalfa, BW was increased in 5th cutting-supplemented mice by 0.5g ± 0.2g (2.6%) and 0.4g ± 0.2g (2.0%) at 5 and 6dpi, regardless of form (*P* = 0.002 and 0.01; Fig 4A). In the later timepoints of the infection (14-21dpi), the main effects of both aqueous and chloroform extracts positively impacted BW (Fig 4B and 4C). At 15dpi, mice fed aqueous and chloroform extracts weighed 1.4g ± 0.7g (7.1%) and 1.9g ± 0.7g (9.5%) more than mice fed hay, respectively, regardless of cutting (*P* = 0.03; Fig 4B).

In examining recovery post-inoculation, mice fed control diets recovered to pre-infection BW (18.9g) at 4dpi but experienced additional losses in BW. Full recovery to pre-infection BW in mice fed the control diet occurred at 13dpi followed by an increase to a final BW of 19.7g (4.1% increase over pre-infection; Fig 5). Mice fed 5th cutting chloroform extract displayed an expedited recovery to pre-infection BW at 2dpi, followed by a BW that stayed numerically higher than the control until 9dpi (Fig 5F). For 4 days in the middle of the infection (10- 13dpi),

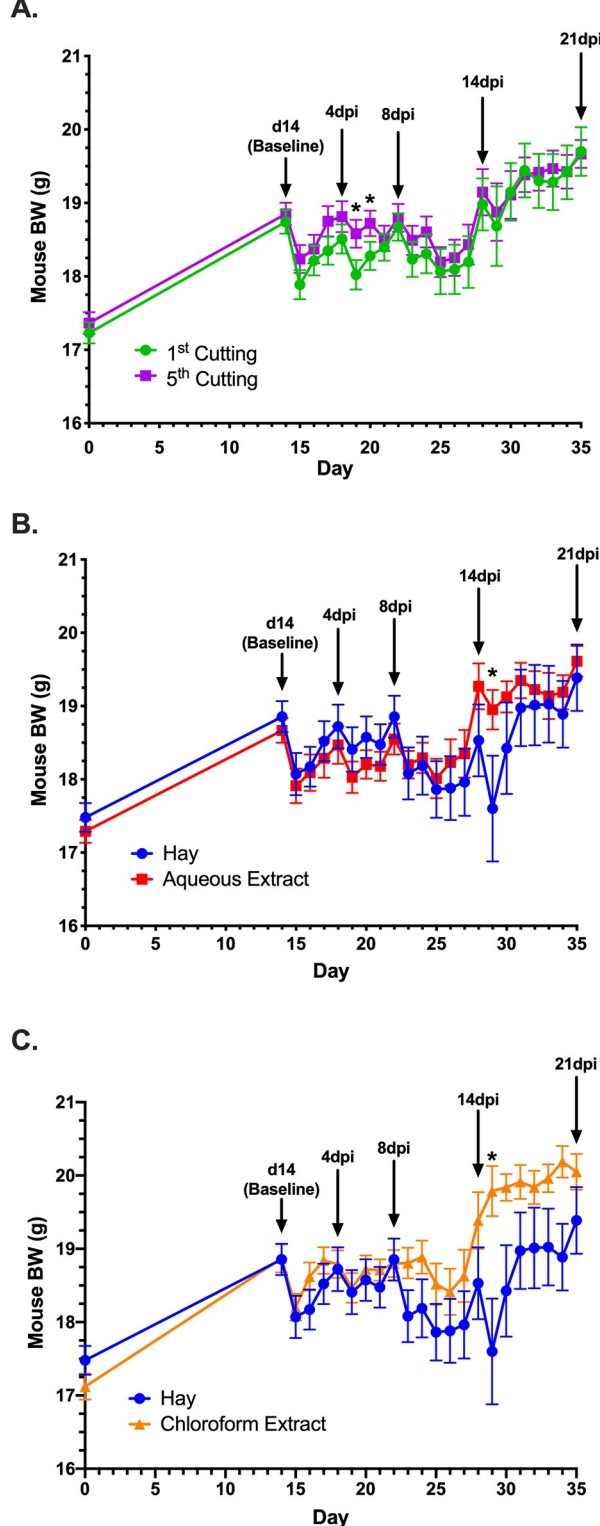

**Fig 4. Isolated BW comparisons of significant BW differences due to the main effects of cutting or supplementation form.** (**A**) 1st vs. 5th cutting alfalfa regardless of alfalfa supplementation form, (**B**) hay vs. aqueous extract and (**C**) hay vs. chloroform extract of alfalfa, regardless of cutting. Key timepoints for blood and tissue sampling are marked by black arrows. Data represent the mean ± SEM, * = P ≤ 0.05.

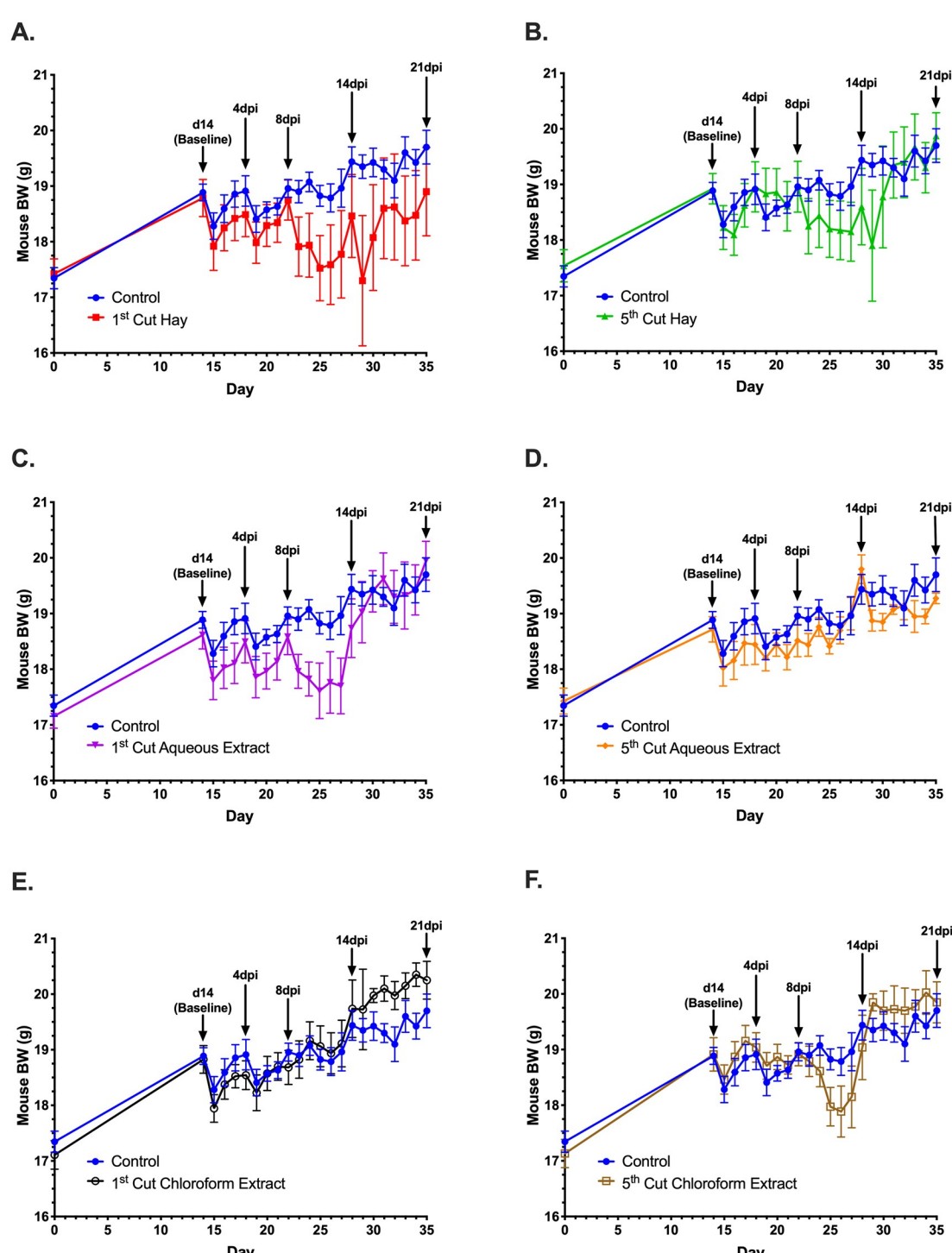

**Fig 5. Isolated comparisons of each alfalfa-supplemented treatment vs. the unsupplemented control.** The control diet is represented as the blue line in each figure. Data represent the mean BW ± SEM. Key timepoints for blood and tissue sampling are marked by black arrows.

the BW of mice fed 5th cutting chloroform extracts dropped to 1g (5.3%) below the pre-infection BW before achieving a final recovery at 14dpi and increasing to a final BW 0.2g (1.0%) above the control. In contrast, mice fed diets with 1st cutting chloroform extract did not show recovery to pre-infection BW until 9dpi but did not experience BW loss following recovery and increased to a final BW that was 0.6g (2.7%) greater than control. Notably, mice fed the chloroform extracts from either cutting had final BW that were greater than the control (Fig 5E and 5F).

## Cytokine-producing cells

The observed patterns in BW and ADFI suggested that chloroform extracts had the potential to improve these parameters as general health indicators during *C. rodentium* infection. Analysis of the immune system and intestinal microbiota in these treatments was conducted in order to investigate some of the underlying physiologies contributing to these observations. Pro-inflammatory interferon (IFN)-γ is a cytokine produced by a number of cell types including antigen-presenting cells and both CD4+ and CD8+ T-cell subpopulations [47]. In healthy mice at d14, chloroform extracts from 1st cutting alfalfa increased the percentage of IFN-γ-producing cells over the control by 14.7%, respectively ($P = 0.002$; Fig 6A). Over the course of infection with *C. rodentium*, mice fed 1st and 5th cutting chloroform extracts displayed 50.0 and 28.2% reductions in IFN-γ+ cells from 0-4dpi to levels 44.9 and 37.9% below the control, respectively ($P < 0.0001$). At 14dpi, splenic populations of IFN-γ-producing cells in mice fed the control diet were reduced by 51.6% to levels 28.8 and 26.2% below 1st and 5th cutting

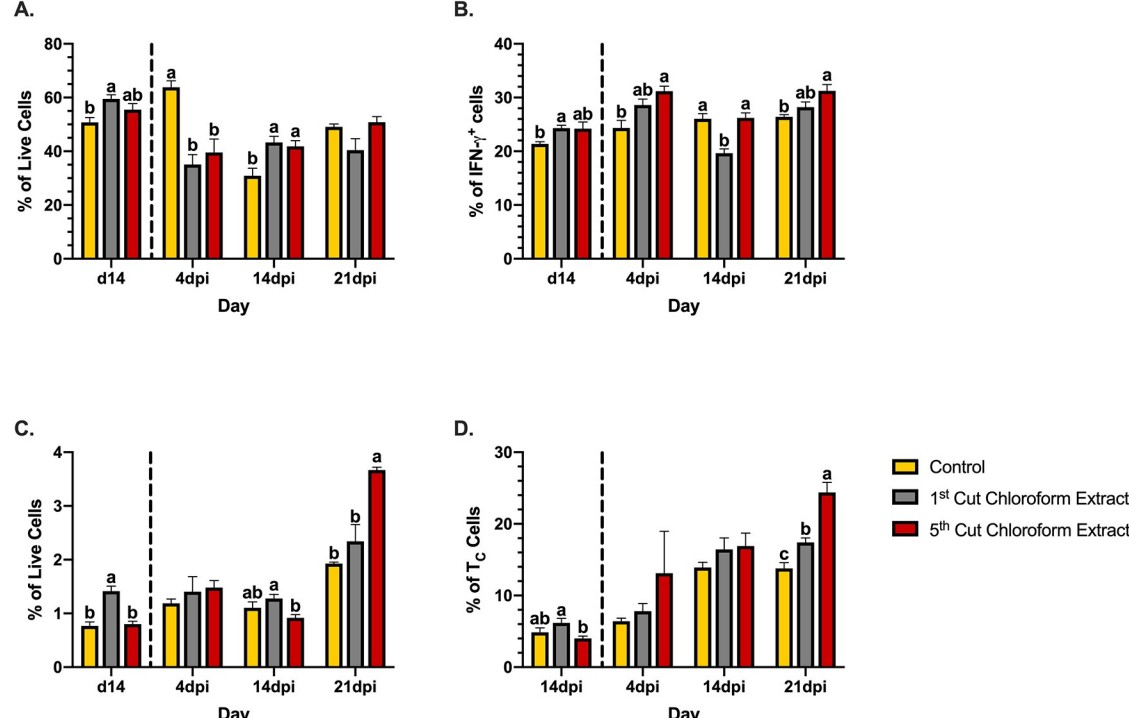

**Fig 6. Splenic cytokine-producing cells in mice fed diets ± 1st or 5th cutting chloroform extract before and after *Citrobacter rodentium* inoculation.** (A) IFN-γ-producing cells, (B) IFN-γ+CD4+ T-helper 1 (TH1) cells, (C) TNF-α-producing cells, and (D) TNF-α-producing CD3+CD8+ T-cytotoxic (TC) cells in the murine spleen. Data are represented as the mean percentage of each cell type ± SEM. Bars with different superscripts are significantly different at $P \leq 0.05$. The dashed line separates the end of the feeding-enrichment period and the start of the infection period.

chloroform extracts, respectively ($P$ = 0.003). Notably, at this point in the infection, splenic IFN-γ⁺ cells increased by 18.9% in diets supplemented with 1st cutting chloroform extract but remained constant in mice fed diets with 5th cutting extract. While no differences between each treatment were observed in the last infection timepoint (21dpi), mice fed 1st cutting chloroform extracts had splenic percentages of IFN-γ⁺ cells 32.1% below pre-infection levels while both the control and 5th cutting chloroform extract diets showed recovery in this cell population (Fig 6A).

Co-expression of IFN-γ with CD4 or CD8 was measured to identify splenic T-cell populations underlying the observed IFN-γ responses. Of the measured IFN-γ⁺ cells, T-helper 1 (T$_H$1; IFN-γ⁺CD4⁺) cells comprised 20–35% of cytokine production at a baseline health state (Fig 6B), while IFN-γ⁺CD8⁺ cells accounted for a lower percentage (4–7%; S1A Fig). In healthy mice, 1st cutting chloroform extract-supplemented diets increased T$_H$1 cells by 12.1% compared to control ($P$ < 0.0001). Throughout the infection, the percentage of splenic T$_H$1 cells in the control diet remained relatively constant, while shifts in these populations were observed in both chloroform extract diets. From 0-4dpi, mice fed 5th cutting chloroform extract-diets showed increases in T$_H$1 cells by 22.3% to levels 21.9% above the control ($P$ = 0.0009). At 14dpi, splenic T$_H$1 cells decreased by 31.3% in mice fed 1st cutting chloroform extract to levels approximately 25% less than both the control and 5th cutting chloroform extract diets ($P$ < 0.0001). In the final days of the infection from 14-21dpi, T$_H$1 cells increased in both 1st and 5th cutting chloroform extract diets by 30.4 and 16.1%, respectively. At 21dpi, all treatments showed percentages of T$_H$1 slightly elevated above pre-infection levels with 5th cutting chloroform extracts having 15.4% greater percentages of this cell type compared to the control ($P$ = 0.001; Fig 6B).

Another pro-inflammatory cytokine, TNF-α, was also measured using intracellular cytokine staining. This cytokine is produced by both innate and adaptive immune cell types [48–50]. Compared to IFN-γ, the percentage of TNF-α⁺ cells in the spleens of healthy mice was considerably lower (~1% compared to ~50–60%; Fig 6C). Diets supplemented with 1st cutting chloroform extract had the greatest percentage of TNF-α⁺ cells at d14 with levels 45.8 and 43.5% above both the control and 5th cutting chloroform extract diets ($P$ < 0.0001). From 4-14dpi, while control and 1st cutting chloroform extract diets maintained TNF-α⁺ populations, 5th cutting chloroform extracts displayed a 38.0% reduction in these cells to levels 28.1% below the 1st cutting chloroform extract diet ($P$ = 0.003). In the last phase of the infection from 14-21dpi, all treatments showed increases in the percentage of TNF-α⁺ cells, with the greatest increase observed in 5th cutting chloroform extracts (75%) to levels 47.4 and 36.2% above both the control and 1st cutting chloroform extract diets ($P$ < 0.0001; Fig 6C).

Production of TNF-α in two different cell types was conducted to determine which populations were responsible for the observed changes: macrophages and CD3⁺CD8⁺ T-cytotoxic (T$_C$) cells. Production of TNF-α is often associated with macrophage activity, among other outcomes; however, only a small percentage of splenic macrophages stained positively for TNF-α (1–2%) and showed consistent changes over the course of infection across all treatments (S1B Fig). At a baseline health state, TNF-α⁺ staining was observed in approximately 5% of T$_C$ cells with 5th cutting chloroform extracts having 35% lower percentages of this cell type compared to 1st cutting chloroform extracts ($P$ = 0.01). Notably, percentages of TNF-α⁺ T$_C$ cells increased in the spleen at later timepoints during the infection in a pattern similar to overall TNF-α⁺ cells (Fig 6D). At 21dpi, 1st and 5th cutting chloroform extracts had 43.5 and 20.8% more TNF-α⁺ T$_C$ cells than the control, respectively, with 5th cutting chloroform extracts having 28.6% more of these cells than 1st cutting chloroform extract ($P$ < 0.0001).

## Innate immune cells

Two innate immune cells measured in the spleen were neutrophils and macrophages based on the expression of CD11b (S1C Fig) with other extracellular markers. Neutrophils (CD11b$^+$Ly6G$^+$) were detected in the spleens of healthy mice at low percentages (3–4%). Healthy mice (d14) mice fed 5$^{th}$ cutting chloroform extracts had 28.0 and 20.3% fewer neutrophils than mice fed the control and 1$^{st}$ cutting chloroform extract diets, respectively ($P$ = 0.004; Fig 7A). Throughout the course of infection, mice fed the control and 5$^{th}$ cutting chloroform extract diets did not show notable changes to neutrophil populations, despite observed differences in the percentages of these cells between the two treatments. In the early timepoints of the infection, mice given 1$^{st}$ cutting chloroform extract showed a 49.4% reduction in neutrophils to levels 52.4 and 20.5% below the control and 5$^{th}$ cutting chloroform extract diets, respectively, at 4dpi ($P$ = 0.0002). Mice fed 1$^{st}$ cutting chloroform extract displayed a notable 81.8% increase in neutrophils from 4-14dpi to levels 54.1 and 58.9% above the control and 5$^{th}$ cutting chloroform extract diets, respectively ($P$ < 0.0001). In the final timepoint of the infection, both mice fed the control and 1$^{st}$ cutting chloroform extract diets had levels of splenic neutrophils above preinfection levels and had populations of this cell type approximately 46% above mice fed 5$^{th}$ cutting chloroform extract ($P$ < 0.0001; Fig 7A).

Compared to neutrophils, populations of macrophages (CD11b$^+$Ly6G$^-$F4/80$^+$) in the spleen were greater, with detected populations accounting for approximately 15–17% of all CD11b$^+$ cells with no differences detected between treatments (Fig 7B). Over the course of the infection, mice fed control diets experienced a 66.1% reduction in macrophages at 4dpi. From 0–4 dpi, 1$^{st}$ and 5$^{th}$ cutting chloroform extracts showed a similar decrease in macrophages but to a lesser degree than the control with 42.6 and 43.6% reductions, respectively, to levels 45.2 and 36.5% above the control ($P$ < 0.0001). Mice fed the control diet had a 57.1% increase in macrophage populations from 4-14dpi. During this time, both chloroform extract diets maintained percentages of these cells at levels that were 31.8 and 42.8% below the control, respectively ($P$ = 0.002). From 14-21dpi, control and 1$^{st}$ cutting chloroform diets showed 23.6 and 13.2% reductions in splenic macrophages, whereas 5$^{th}$ cutting chloroform extracts maintained this cell population at approximately 7.5% (23.8% less than control; $P$ = 0.02). None of the treatments displayed a return to pre-infection levels of macrophages by the end of the infection period (Fig 7B).

## Adaptive immune cells

The spleen is characterized as a secondary lymphoid organ and maintains discrete populations of lymphocytes [51]. Approximately 40–50% of the live cells detected in the spleens of healthy

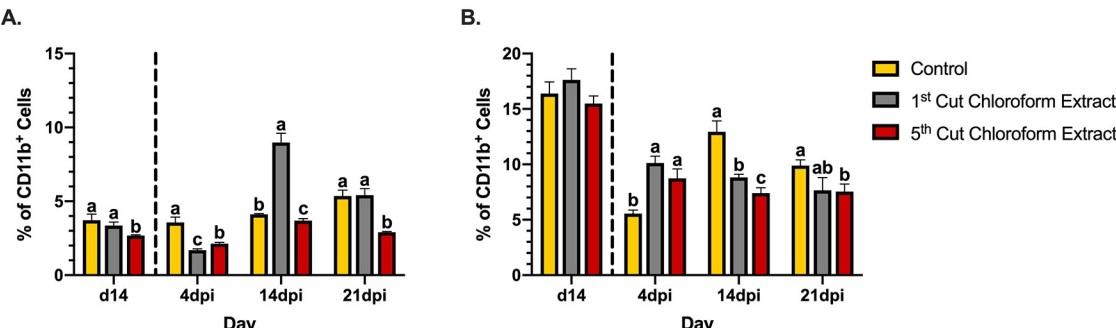

**Fig 7. Splenic innate immune cells in the spleens of mice fed diets ± 1$^{st}$ or 5$^{th}$ cutting chloroform extract before and after** *Citrobacter rodentium* **inoculation.** (A) CD11b$^+$Ly6G$^+$ neutrophils and (B) CD11b$^+$Ly6G$^-$F4/80$^+$ macrophages in the murine spleen. Data are represented as the mean percentage of each cell type ± SEM. Bars with different superscripts are significantly different at $P \leq 0.05$. The dashed line separates the end of the feeding-enrichment period and the start of the infection period.

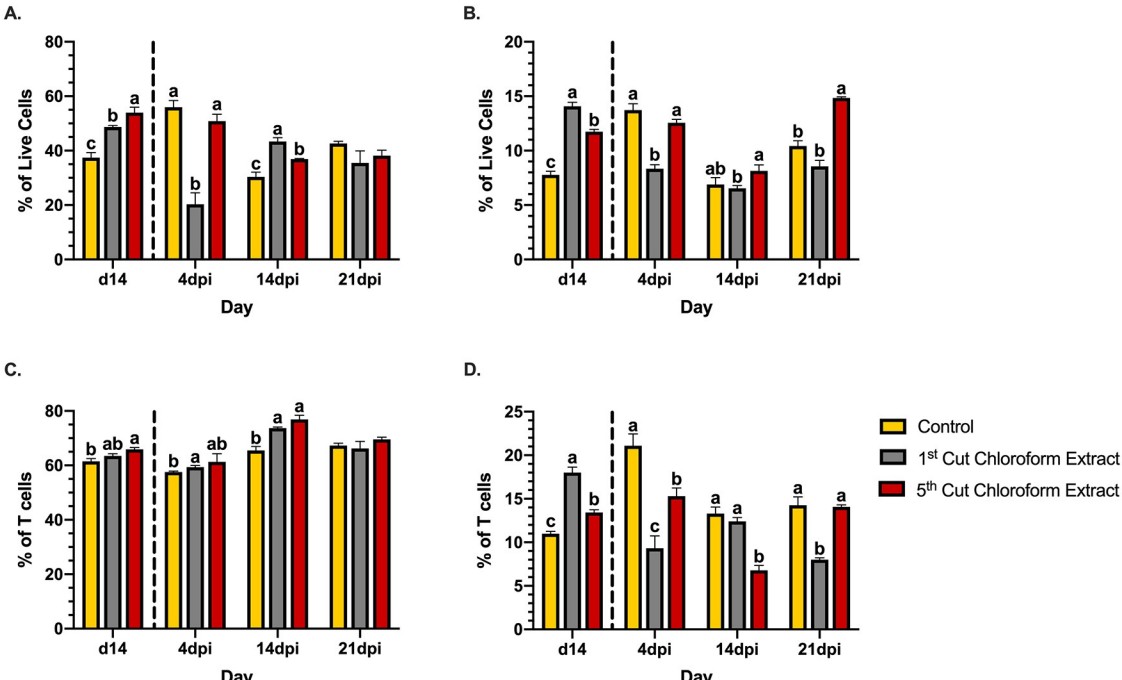

**Fig 8. Splenic adaptive immune cells in mice fed diets ± 1st or 5th cutting chloroform extract before and after *Citrobacter rodentium* inoculation.** (**A**) B220+ B-cells, (**B**) CD3+ T-cells, (**C**) CD3+CD4+ T helper ($T_H$) cells, and (**D**) CD3+CD8+ T-cytotoxic ($T_C$) cells in the murine spleen. Data are represented as the mean percentage of each cell type ± SEM. Bars with different superscripts are significantly different at $P \leq 0.05$. The dashed line separates the end of the feeding-enrichment period and the start of the infection period.

mice were identified as B-cells (B220+; Fig 8). After the 14d feeding enrichment period, mice fed both 1st and 5th cutting chloroform extracts had 23.2 and 30.7% more B-cells than the control, respectively ($P < 0.0001$; Fig 8A). At 4dpi, control diets showed a 33.2% increase in B-cells, while 5th cutting chloroform extracts maintained splenic B-cell populations. Most notably, 1st cutting chloroform extracts showed a 58.2% reduction in B-cells from 0-4dpi to levels 63.7 and 60.0% below the control and 5th cutting chloroform extract diets, respectively ($P < 0.0001$). At 14dpi, B-cell populations in the spleens of mice fed the control diet were reduced by 45.7% to levels below both chloroform extracts ($P < 0.0001$). Mice fed 5th cutting chloroform extracts showed a similar decrease in splenic B-cells from 4-14dpi, but to a lesser degree than observed in the control diet (27.4%). In contrast, mice fed 1st cutting chloroform extract displayed a 53.1% increase in splenic B-cells at this timepoint. In the final timepoint of infection, B-cell populations remained below pre-infection levels in the spleens of mice fed both chloroform extracts, while the control diet showed recovery to pre-infection levels (Fig 8A).

Changes to overall populations of CD3+ T-cells were similar to those observed in B-cells. At a baseline health state, both chloroform extracts increased splenic T-cell populations compared to control, with 1st cutting chloroform extracts having the greatest T-cell presence at 14.1% ($P < 0.0001$; Fig 8B). At 4dpi, mice fed the control diet had a 43.4% increase in the percentage of T-cells, while those fed 5th cutting chloroform extract maintained populations of these cells. Similar to patterns observed in B-cells, 1st cutting chloroform extract diets showed early recruitment of T-cells characterized by a 40.7% reduction to levels below both the control and 5th cutting alfalfa diets ($P < 0.0001$). At 14dpi, mice fed the control diets experienced a 49.9% reduction in T-cells while those fed 5th cutting chloroform extract showed a 35.1% reduction

in this population. Mice fed 1st cutting alfalfa showed further reductions to levels maintained below 5th cutting chloroform extract ($P = 0.04$). In the last day of the infection period (21dpi), all treatments showed increases in T-cell populations, but only 1st cutting chloroform extract diets did not display recovery to T-cell populations above pre-infection levels and remained lower than those seen in mice fed 5th cutting chloroform extract ($P < 0.0001$; Fig 8B).

T-cell populations were further divided into CD3$^+$CD4$^+$ T$_H$ and CD3$^+$CD8$^+$ T$_C$ subpopulations to identify which were responding to alfalfa supplementation and *C. rodentium* infection. Notable differences in T$_H$ populations between treatments were not observed after the feeding enrichment period and changes over the course of the infection were consistent across treatments (Fig 8C). In contrast, changes to T$_C$ cells roughly corresponded to changes in overall T-cell populations (Fig 8D).

## Colon microbiota

In terms of whole community comparisons, 1st cutting chloroform extracts did not differ from control as determined by ANOSIM. Feeding diets supplemented with 5th cutting chloroform extract altered the intestinal microbiota at whole-community levels compared to control and 1st cutting chloroform extract at both a baseline health state ($P = 0.04$ and 0.03, respectively) and 4dpi ($P = 0.03$). Chloroform extracts of 5th cutting alfalfa did not alter the microbiome at a whole-community level at later timepoints of *C. rodentium* infection (14 and 21dpi; Table 2). Principal coordinates analysis (PCoA) showed variations in the clustering pattern of different dietary treatments throughout the course of the study, with 5th cutting chloroform extracts tending to cluster differently from both the control and 1st cutting chloroform extracts (Fig 9). In addition, a clustering of samples according to sampling days was observed indicating that the microbial communities changed over time independent of the dietary additives.

**Table 2. Whole-community ANOSIM comparisons of the colon microbiota of mice fed the control or diets supplemented with chloroform extracts from 1st and 5th cutting alfalfa.**

| Comparison | R-value[2] | P-value[3] |
|---|---|---|
| *Control vs. 1st Cut Chloroform Extract* | | |
| d14 (baseline) | 0.18 | 0.09 |
| 4dpi | 0.39 | 0.08 |
| 14dpi | 0.06 | 0.29 |
| 21dpi | 0.00 | 0.52 |
| *Control vs. 5th Cut Chloroform Extract* | | |
| d14 (baseline) | 0.24 | 0.04 |
| 4dpi | 1.00 | 0.03 |
| 14dpi | -0.13 | 0.80 |
| 21dpi | 0.23 | 0.11 |
| *1st Cut Chloroform vs. 5th Cut Chloroform Extract* | | |
| d14 (baseline) | 0.42 | 0.03 |
| 4dpi | 0.54 | 0.03 |
| 14dpi | 0.14 | 0.17 |
| 21dpi | -0.03 | 0.54 |

[1] Analysis of similarity performed using mothur.

[2] R-values detail the source of sample variations on a scale of -1 to 1. Values closer to -1 suggest higher variation between within samples while those closer to 1 suggest higher variation between samples. R-values close to 0 indicate no differences in variation.

[3] Significance determined at $P \leq 0.05$.

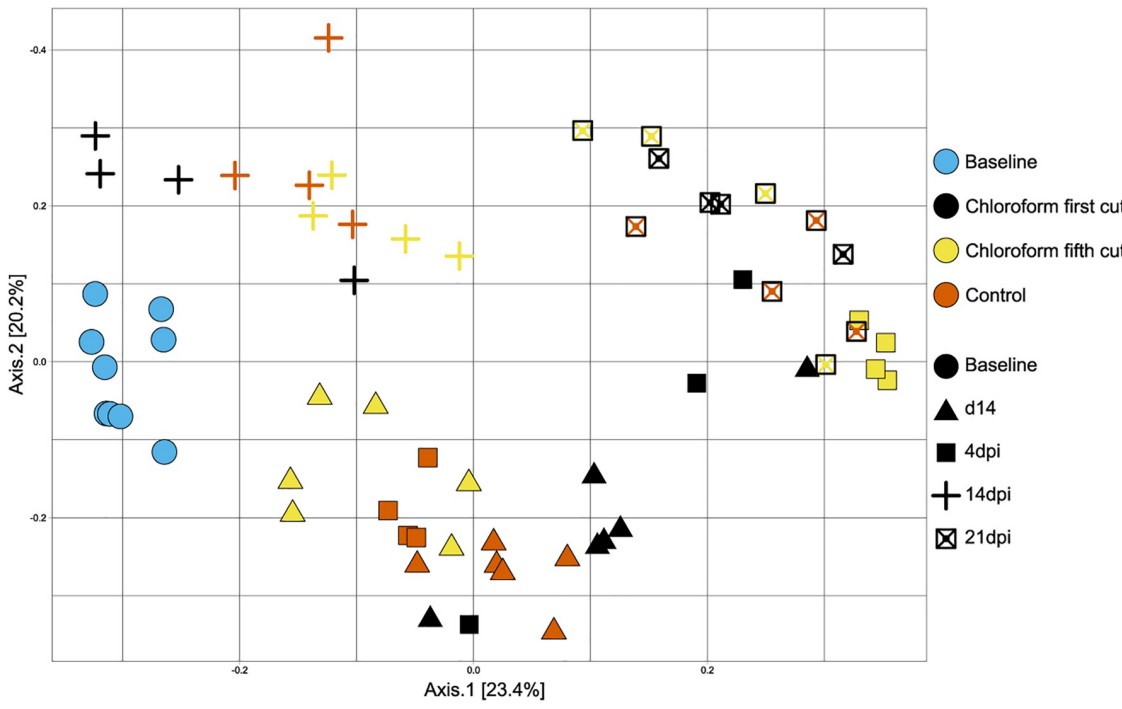

**Fig 9. Beta diversity of colon microbial communities in mice fed diets ± 1st or 5th cutting chloroform extract during *C. rodentium* challenge.** Data are shown for all sampling days of the trial and all plots are based on Bray-Curtis distances.

At the genus level, OTUs associated with *Muribaculaceae* and *Lachnospiraceae* families were among the most represented in the colon microbiota throughout the study (Fig 10). *Citrobacter rodentium* was detected in the colon microbiota of inoculated mice and identified as OTU 47. In healthy animals (d14), 5th cutting chloroform extracts resulted in 273 significantly different OTUs compared to control, versus the 164 different OTUs observed between 1st cutting chloroform extract and the control at this timepoint. Of these altered OTUs, 1st cutting chloroform extract reduced the relative abundance of *Faecalibaculum* (OTU 74; *P* = 0.02), while feeding 5th cutting chloroform extract increased the relative abundance of more highly abundant OTUs associated with *Muribaculaceae* (OTUs 1, 14, 26, 34, and 80) out of the 100 most abundant OTUs (S1 and S2 Tables). In particular, 5th cutting chloroform extract increased the relative abundance of *Muribaculaceae* OTUs 1, 26, 34, and 80 over both the

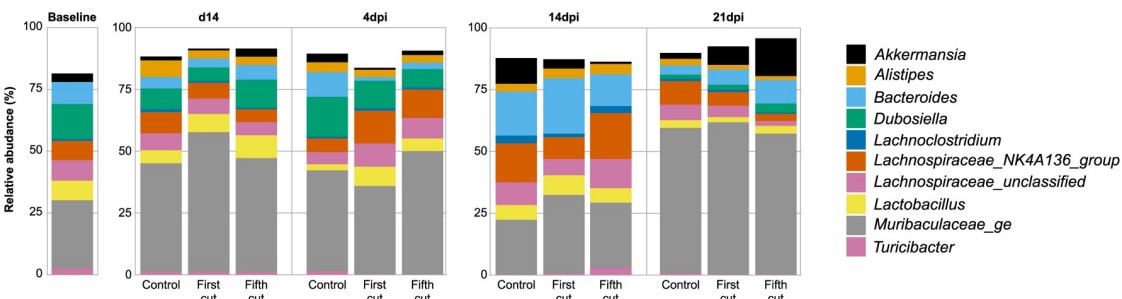

**Fig 10. Relative abundance of the 10 most abundant genera in the colon microbiota of healthy and *Citrobacter rodentium*-challenged mice fed diets ± 1st or 5th cutting chloroform extract.**

control and 1st cutting chloroform diets, which may be contributing to the significantly altered overall community (S2 and S3 Tables).

In the earliest infection timepoint (4dpi), feeding 1st cutting chloroform extract resulted in 157 different OTUs from the control whereas 5th cutting extract resulted in 553 significantly different OTUs from the control. These changes are likely due to the greater relative abundance of 13 OTUs associated with *Muribaculaceae* and three OTUs associated with *Lachnospiraceae* in mice fed 5th cutting chloroform extracts (S5 Table). In contrast, feeding 1st cutting chloroform extract increased 7 OTUs association with *Lachnospiraceae* and only 1 associated with *Muribaculaceae*, while also increasing the relative abundance of the genera *Lactobacillus* (OTU 6; $P = 0.03$) while reducing the relative abundance of *Roseburia* (OTU 58; $P = 0.03$; S4 Table). In addition to increasing OTUs associated with *Muribaculaceae*, 5th cutting chloroform extract reduced the relative abundance of *Parasutterella* (OTU 21; $P = 0.02$) and *Bifidobacterium* (OTU 19) compared to control ($P = 0.02$; S4 Table). The most notable change observed at 4dpi was the significant reduction in the relative abundance of *C. rodentium* (OTU 47) in the colon of mice fed 5th cutting chloroform extracts compared to the control ($P = 0.02$; Fig 11; S5 Table).

Compared to the control, the number of significantly different OTUs between both chloroform extracts did not vary greatly at 14dpi, with 291 different OTUs in mice fed 1st cutting chloroform extract and 261 in those fed 5th cutting extract. Both chloroform extracts increased the relative abundance of *Turicibacter* (OTU 13) compared to control ($P = 0.02$), with 5th cutting chloroform extracts having 4.9-fold greater abundance of this OTU compared to 1st cutting ($P = 0.02$; Fig 11; S7 and S8 Tables). Additionally, feeding 1st cutting chloroform extract increased the relative abundance of *Bifidobacterium* (OTU 19; $P = 0.02$) and *Romboutsia* (OTU 49) compared to both the control and 5th cutting extracts ($P = 0.04$ and 0.02, respectively; S7 and S9 Tables). Similar to observations at 4dpi, feeding 5th cutting chloroform extract had greater impacts on highly abundant OTUs associated with *Muribaculaceae* at this timepoint, with increases to OTUs 4, 45, 57, and 62 compared to control and 1st cutting (OTUs 4, 57, and 62 only; S8 and S9 Tables). Feeding 5th cutting chloroform also decreased the relative abundance of *Roseburia* (OTU 58) compared to control and 1st cutting extract ($P = 0.01$ and 0.05, respectively).

At the conclusion of the study, corresponding with *C. rodentium* resolution (21dpi), mice fed 1st cutting chloroform extracts had 263 significantly different OTUs compared to the control whereas mice fed 5th cutting extracts had 466 different OTUs. At this timepoint, mice fed 1st cutting chloroform extract had increased relative abundance of *Roseburia* (OTU 58) compared to control and 5th cutting (P = 0.04), with additional increases in *Mollicutes* (OTU 67) compared to control ($P = 0.05$; S10 and S12 Tables). Feeding 5th cutting chloroform extract increased the relative abundance of *Akkermansia* (OTU 8; Fig 11) compared to the control ($P = 0.02$) along with reducing the relative abundance of *Oscillibacter* (OTU 36) compared to control and 1st cutting *(P = 0.02$; S11 and S12 Tables).

## Discussion/Conclusions

Body weight and ADFI outcomes were used to focus further investigation into underlying changes to immunity and the colon microbiota in response to the varying forms and cuttings of alfalfa supplemented. In livestock, performance measurements like BW gain and ADFI are commonplace, but these measures are rarely reported in mouse models due to differences in husbandry objectives and difficulties in accurately measuring rodent FI. While not typical for the model used here, results from the BW and ADFI measurements showed no differences in ADFI and BW due to alfalfa supplementation in healthy mice; however, post-inoculation responses varied considerably based on diet. The consistently greater ADFI observed in mice

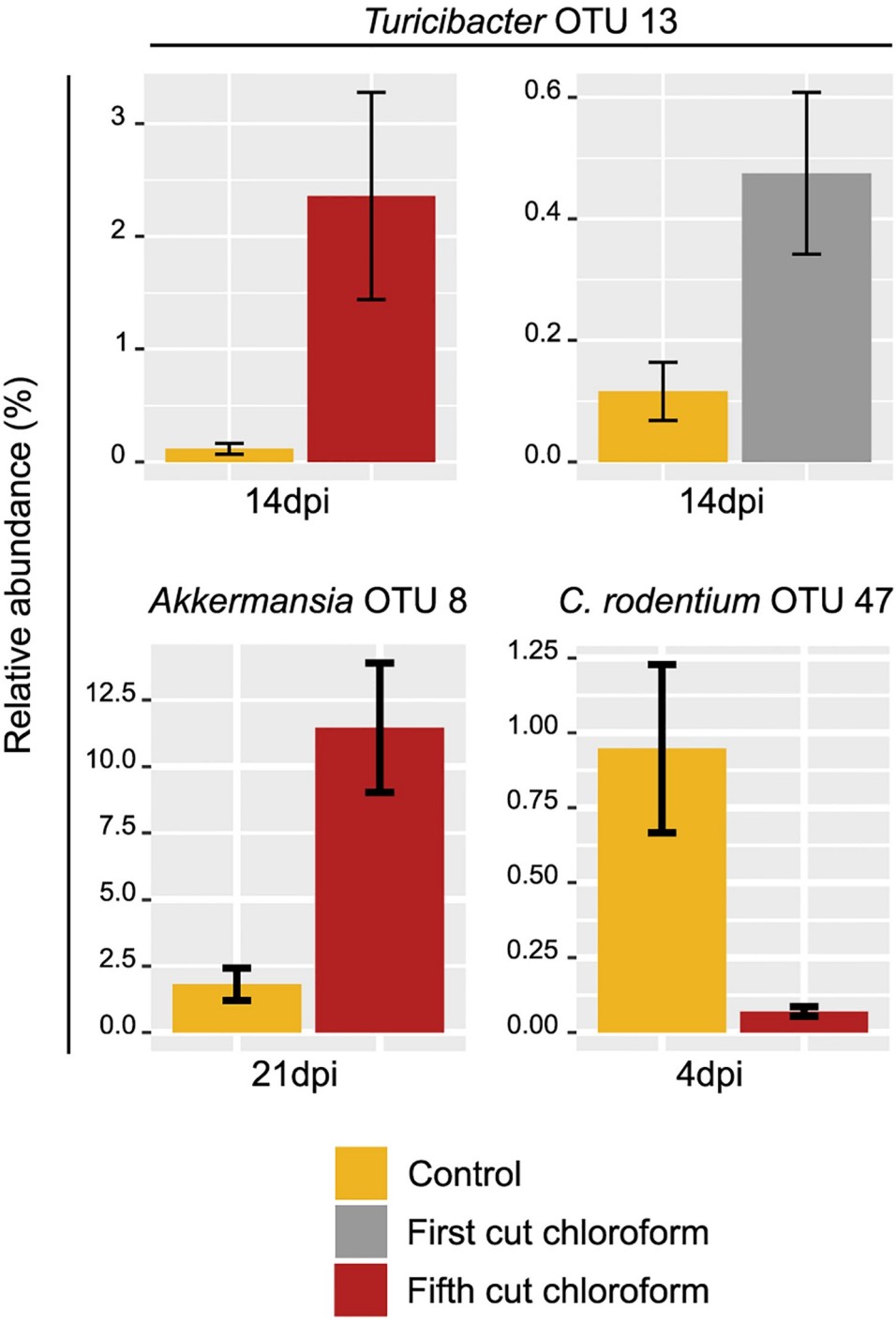

**Fig 11. Relative abundances of select OTUs during different timepoints of *Citrobacter rodentium* infection in mice fed control, 1st cutting chloroform extract, and 5th cutting chloroform extract diets.**

fed hay diets was likely a function of increased dietary fiber reducing dietary energy availability causing mice to increase ADFI to fulfill energy requirements while post-inoculation BW results suggest an inability to adequately compensate during *C. rodentium* infection (Fig 5A and 5B) [52]. Observed responses throughout the infection period suggest that late-cutting alfalfa had a

protective effect on BW in the early timepoints of *C. rodentium* infection, while supplementation form, particularly chloroform extracts, had greater impacts at later timepoints. Notably, both outcomes were observed in mice fed 5th cutting chloroform extracts, suggesting that enriched lipid-soluble phytochemicals in the later cutting may have a protective effect on mouse BW during health challenge. These observations led to in-depth analysis into how underlying changes to the immune system and microbiota in mice fed chloroform extracts may be contributing to the observed BW responses before and after infection.

Healthy mice fed chloroform extracts from both 1st and 5th cutting alfalfa displayed shifts in splenic immune cell profiles and colon microbiota. After 14d of feeding enrichment, 1st cutting chloroform extracts contribute to a pro-inflammatory environment as evidenced by the significant increases in both IFN-γ and TNF-α-producing cells compared to the control diet (Fig 6A and 6C). While 1st cutting chloroform extracts had greater impacts on cytokine-producing cells, extracts from both cuttings increased lymphocyte populations similar to observations reported in non-ruminant livestock [13, 16, 17]. At this timepoint, the microbiota of mice fed 5th cutting extracts differed from both the control and 1st cutting extracts, likely due to changes in the relative abundance of highly represented *Muribaculaceae* OTUs. While these changes were determined to be significant, similarities in BW and ADFI at d14 indicate that alterations to the immune system and microbiota were not substantial enough to negatively impact these indicators of general health.

Implementation of a rodent-specific challenge with *C. rodentium* provided functional insights into physiological changes as they translated to general health responses. *Citrobacter rodentium* is often limited to the colon and is not known to colonize the spleen, which limits our understanding of the spleen's role during infection [39]. Given the spleen's role in initiating innate and adaptive immune responses to infection, reductions in splenic cell populations may indicate recruitment to sites of infection in the peripheral tissues [51]. Infection with *C. rodentium* is characterized by a strong IFN-γ and its associated $T_H1$ response in addition to dysbiosis as pathogen overgrowth displaces the resident microbiota [37, 53]. *Citrobacter rodentium* was not present in the colons of healthy mice and was identified at 4dpi before returning to low or undetectable levels by the later infection timepoints. Both chloroform extracts displayed early recruitment of IFN-γ-producing cells to peripheral tissues as evidenced by reductions in splenic populations at 4dpi versus 14dpi in the control diet (Fig 6A). Although changes in $T_H1$ populations were observed in the spleen (Fig 6B), overall $T_H$ populations did not display evidence of recruitment to the site of infection (Fig 8C). Together, these results suggest that recruitment of IFN-γ-producing cells to peripheral tissues were not associated with a $T_H1$ response. Instead, the observed fluctuations of splenic $T_H1$ cells were likely a function of shifts in overall IFN-γ+ populations and any $T_H1$ activity during infection was localized to the colon.

During the early timepoints post-inoculation, mice fed 5th cutting chloroform extracts rapidly recovered to their pre-infection BW within 2d of inoculation, and some of the underlying physiological responses may have contributed to this response. In terms of immunity, 5th cutting chloroform extracts maintained elevated levels of lymphocyte populations at 4dpi while control diets showed evidence of rapidly expanding cell populations and 1st cutting chloroform extracts exhibited early recruitment of lymphocytes to peripheral tissues 10d before expected. Additionally, 5th cutting chloroform extracts significantly reduced the presence of *C. rodentium* in the colon. Combined, these results suggest that 5th cutting chloroform extracts reduced the ability of *C. rodentium* to colonize the murine colon and did not contribute to potentially detrimental early recruitment of adaptive immune cells, which may have contributed to the observed BW responses.

During resolution of *C. rodentium* infection at later timepoints both chloroform extracts maintained reduced splenic macrophage populations while control diets showed fluctuations in

this cell type in later timepoints (Fig 7B). Similarly, both chloroform extracts showed signs of prolonged B-cell responses compared to the control. While splenic B-cells and macrophages did not show recovery to pre-inoculation levels at later timepoints in mice fed both chloroform extracts, both extracts notably increased the relative abundance of potentially beneficial *Turicibacter* at 14dpi and only 5th cutting extracts increased *Akkermansia* at 21dpi (Fig 11). Increased relative abundance of *Turicibacter* in chickens has been associated with low residual feed intake (RFI), which roughly corresponds with observations that mice fed chloroform extract-supplemented diets ate the lowest amount of feed and weighed more than mice fed the control diet in later timepoints post-inoculation [54]. This suggests that increased *Turicibacter* abundance may contribute to a similar phenotype in different species; however, the exact similarities remain unclear as RFI is not typically assessed in mouse models. *Akkermansia* is a genus that is inversely correlated with inflammatory bowel disease and generally regarded as anti-inflammatory [55]. In the final days of *C. rodentium* infection, both chloroform extracts resulted in numerically greater BW compared to the control, which may be the result of prolonged immune cell responses, increases in beneficial microbial genera, or a combination of the two.

The identity of compounds that may be responsible for these effects remains unclear as utilization of chloroform alfalfa extracts in animal research is limited, with one study examining the effects of chloroform alfalfa extract to date. Work published by Choi and colleagues (2013), found that chloroform alfalfa extract reduced *in vitro* pro-inflammatory cytokine production during LPS challenge and suggested that palmitic, linoleic, linolenic, and a number of phenolic acids may be causing the observed effects [18]. The structural similarity of alfalfa-derived phytoestrogens to lipid-soluble steroid hormones, suggests that they may also be present in lipid-soluble chloroform extracts; however, their presence was identified in aqueous extract used by Dong and colleagues (2007) and isoflavones are generally regarded as being water-soluble [16, 56]. The lack of published literature concerning chloroform alfalfa extracts and the fact that this study did not specifically characterize the phytochemicals present in chloroform extract leaves the identity of these compounds largely unknown.

The concurrent assessment of the immune system and microbial communities of both healthy and pathogen-challenged mice provided insights into changes underlying general measurements of BW and ADFI; however, the assays used to analyze these systems do not provide specific functional insights. While measuring spleen immune cell profiles showed systemic responses to *C. rodentium*, site-specific responses to *C. rodentium* were difficult to measure due to the low yield of immune cells obtained from the colon. Despite these limitations, the results obtained by this work demonstrate that lipid-soluble compounds present in alfalfa modulate host immunity and the intestinal microbiota to improve BW in the late stages of an immune challenge, while those derived from late-cutting alfalfa have greater impacts immediately following inoculation. These insights gained from well-characterized mouse model can guide future livestock applications to focus more specifically on the combined benefits of lipid-soluble compounds from late-cutting alfalfa in production animals.

## Supporting information

**S1 Fig. Percentage of select immune cell populations in the spleens of mice basal diet ± 1st or 5th cutting chloroform extracts.** (A) IFN-γ+CD8+ cytotoxic T cells, (B) TNF-α+ macrophages, and (C) CD11b+ cells. Data are represented as the mean percentage of each cell type ± SEM. Bars with different superscripts are significantly different at $P \leq 0.05$. The dashed line separates the end of the feeding-enrichment period and the start of the infection period. (TIFF)

**S1 Table. Significantly different OTUs in the colon microbiota of healthy mice fed the control diet vs. 1st cutting chloroform extract at d14.**
(PDF)

**S2 Table. Significantly different OTUs in the colon microbiota of healthy mice fed the control diet vs. 5th cutting chloroform extract at d14.**
(PDF)

**S3 Table. Significantly different OTUs in the colon microbiota of healthy mice fed 1st cutting chloroform extract vs. 5th cutting chloroform extract at d14.**
(PDF)

**S4 Table. Significantly different OTUs in the colon microbiota of healthy mice fed the control diet vs. 1st cutting chloroform extract at 4dpi.**
(PDF)

**S5 Table. Significantly different OTUs in the colon microbiota of healthy mice fed the control diet vs. 5th cutting chloroform extract at 4dpi.**
(PDF)

**S6 Table. Significantly different OTUs in the colon microbiota of healthy mice fed 1st cutting chloroform extract vs. 5th cutting chloroform extract at 4dpi.**
(PDF)

**S7 Table. Significantly different OTUs in the colon microbiota of healthy mice fed the control diet vs. 1st cutting chloroform extract at 14dpi.**
(PDF)

**S8 Table. Significantly different OTUs in the colon microbiota of healthy mice fed the control diet vs. 5th cutting chloroform extract at 14dpi.**
(PDF)

**S9 Table. Significantly different OTUs in the colon microbiota of healthy mice fed 1st cutting chloroform extract vs. 5th cutting chloroform extract at 14dpi.**
(PDF)

**S10 Table. Significantly different OTUs in the colon microbiota of healthy mice fed the control diet vs. 1st cutting chloroform extract at 21dpi.**
(PDF)

**S11 Table. Significantly different OTUs in the colon microbiota of healthy mice fed the control diet vs. 5th cutting chloroform extract at 21dpi.**
(PDF)

**S12 Table. Significantly different OTUs in the colon microbiota of healthy mice fed 1st cutting chloroform extract vs. 5th cutting chloroform extract at 21dpi.**
(PDF)

**S1 Data.**
(XLSX)

## Author Contributions

**Conceptualization:** S. Schmitz-Esser, E. A. Bobeck.

**Data curation:** J. M. Anast, S. Schmitz-Esser.

**Formal analysis:** K. Fries-Craft, J. M. Anast.

**Funding acquisition:** S. Schmitz-Esser, E. A. Bobeck.

**Investigation:** K. Fries-Craft, J. M. Anast, S. Schmitz-Esser, E. A. Bobeck.

**Methodology:** E. A. Bobeck.

**Project administration:** K. Fries-Craft, E. A. Bobeck.

**Resources:** K. Fries-Craft, J. M. Anast, S. Schmitz-Esser, E. A. Bobeck.

**Software:** J. M. Anast.

**Supervision:** S. Schmitz-Esser, E. A. Bobeck.

**Visualization:** K. Fries-Craft, J. M. Anast, S. Schmitz-Esser, E. A. Bobeck.

**Writing – original draft:** K. Fries-Craft.

**Writing – review & editing:** J. M. Anast, S. Schmitz-Esser, E. A. Bobeck.

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
