## [Decision Letter · Decision Letter 0]

30 Apr 2020

PONE-D-20-06132

Host immunity and the colon microbiota of mice infected with Citrobacter rodentium are beneficially modulated by lipid-soluble compounds from late-cutting alfalfa in the early stages of infection

PLOS ONE

Dear Dr Bobeck,

Thank you for submitting your manuscript to PLOS ONE. After careful consideration, we feel that it has merit but does not fully meet PLOS ONE’s publication criteria as it currently stands. Therefore, we invite you to submit a revised version of the manuscript that addresses the points raised during the review process.

We would appreciate receiving your revised manuscript by Jun 14 2020 11:59PM. To enhance the reproducibility of your results, we recommend that if applicable you deposit your laboratory protocols in protocols.io, where a protocol can be assigned its own identifier (DOI) such that it can be cited independently in the future. For instructions see: http://journals.plos.org/plosone/s/submission-guidelines#loc-laboratory-protocols

We look forward to receiving your revised manuscript.

Kind regards,

Juan J Loor

Academic Editor

PLOS ONE

Reviewers' comments:

Reviewer's Responses to Questions

**Comments to the Author**

1. Is the manuscript technically sound, and do the data support the conclusions?

Reviewer #1: Yes

2. Has the statistical analysis been performed appropriately and rigorously? 

Reviewer #1: Yes

3. Have the authors made all data underlying the findings in their manuscript fully available?

Reviewer #1: No

4. Is the manuscript presented in an intelligible fashion and written in standard English?

Reviewer #1: Yes

5. Review Comments to the Author

Reviewer #1: General comments/questions:

-Introduction is extremely long.

-Add a paragraph justifying the choice of inducing an infection with "Citrobacter rodentium" for this study

- How did you demonstrate that the challenge with Citrobacter rodentium was effective to study a response?

-I did not see assays that specifically measured the lipid-compounds. So please, edit the title of the manuscript accordingly. From "are beneficially modulated by lipid-soluble compounds from late-cutting alfalfa", to "are modulated by late-cutting alfalfa"

1) Please, reduce the introduction to 1.5 pages max.

2) Focus on explaining the most relevant facts about the use of alfalfa, including:

"documented source of bioactive compounds" - which compounds?; which bacterial species in the gut has been affected upon alfalfa diet? increased commensal bacteria?

Material and methods:

3) Include the Animal# for the IACUC

4) Blood samples were not collected? Why?

5) please, explain which procedure (including the anesthesia) was used for the mouse euthanasia

6) how were the colon contents collected (i.e., separated from the tissue)?

7) After collection, were the spleen samples stored in microtubes containing RMPI-FBS or were immediately frozen?

8) why 6-week old mouse, instead of 14 or 21d old mouse? For instance, wouldn't alfalfa benefit the early life gut colonization?

9) Flow cytometry analysis did not include the aqueous extract treatment? Why?

Results:

10) Please, re word the sentences on lines 253-255, that's confusing

11) the examining recovery method was not explained in material and methods, but it was discussed. Please, add a paragraph explaining the recovery examination post-infection.

12) "Production of TNF-� is often associated with macrophage activity", please delete. TNFa is associated with a variety of activities.

13) Please, correct "(PCoA) " to PCA

Discussion:

14) Please, reduce the discussion about BW. It's too long for 2 parameters.

6. PLOS authors have the option to publish the peer review history of their article (what does this mean?). If published, this will include your full peer review and any attached files.

Reviewer #1: No

---

## [Author Response · Author response to Decision Letter 0]

29 May 2020

It is our pleasure to resubmit our full-length research article titled “Host immunity and the colon microbiota of mice infected with Citrobacter rodentium are beneficially modulated by lipid-soluble extract from late-cutting alfalfa in the early stages of infection” for publication in PLoS ONE [PONE-D-20-06132]. This resubmission is to incorporate edits and address comments made by the reviewer during the peer-review process. The responses to each comment and reference to relevant changes are as follows:

[From Academic Editor] Please ensure that your manuscript meets PLOS ONE's style requirements, including those for file naming. 

Author Response: Line numbering changed to be continuous throughout the manuscript, headings changed to sentence case, and file naming requirements adhered to. 

Reviewer #1: General comments/questions:

-Introduction is extremely long.

Author Response: Introduction shortened to under 2 pages, more detail about these changes are included with more specific recommendations below. 

-Add a paragraph justifying the choice of inducing an infection with "Citrobacter rodentium" for this study

Author response: Justification for the selection of Citrobacter rodentium is given in new lines 75-78. Given that the intro has already been deemed too long, we are comfortable with our existing justification. 

- How did you demonstrate that the challenge with Citrobacter rodentium was effective to study a response?

Author response: The challenge was effectively noted in expected outcomes, including weight loss and feed intake changes noticed immediately in the days after inoculation (Fig 1). The persistence of the organism in control and 1st cutting chloroform extract groups along with differences in weight gain and feed intake responses are indicators that the bacteria had caused the expected gastrointestinal effects on the mice.

-I did not see assays that specifically measured the lipid-compounds. So please, edit the title of the manuscript accordingly. From "are beneficially modulated by lipid-soluble compounds from late-cutting alfalfa", to "are modulated by late-cutting alfalfa"

Author response: These changes were only observed in the lipid-soluble (chloroform) extract of late-cutting alfalfa, removing mention of this is not descriptive of the outcomes. Title changed to reflect that these changes were specific to a lipid-soluble extract and not to specific compounds. 

1) Please, reduce the introduction to 1.5 pages max.

Author Response: As suggested, the introduction has been reduced from 3 pages in the first submission to under 2 pages in length. We feel that the remainder of the introduction is essential for introducing the background of our study and hope that the reviewer will agree.

2) Focus on explaining the most relevant facts about the use of alfalfa, including:

"documented source of bioactive compounds" - which compounds?; which bacterial species in the gut has been affected upon alfalfa diet? increased commensal bacteria?

Author Response: Introduction shortened to compile similar results/outcomes and identify these compounds earlier in the manuscript.

Material and methods:

3) Include the Animal# for the IACUC

Author Response: Protocol number and additional information added to the IACUC statement in accordance with PLoS One reporting requirements (new lines 88-92).

4) Blood samples were not collected? Why?

Author response: Blood samples were taken for serum cytokine analyses by both ELISA and a multi-plex flow cytometric assay. The results from both assays were highly variable/ inconclusive with many samples having undetectable or extremely high levels of serum cytokines. These outcomes were inconsistent across treatments and between the different assays and were deemed insufficient for the report. 

5) please, explain which procedure (including the anesthesia) was used for the mouse euthanasia

Author response: This information was added to new lines 88-89

6) how were the colon contents collected (i.e., separated from the tissue)?

Author Response: Description of method added to new lines 172-173. 

7) After collection, were the spleen samples stored in microtubes containing RMPI-FBS or were immediately frozen?

Author Response: Spleen samples were immediately frozen in cryotubes with RPMI-BCS + 7.5% DMSO. More detail added to new lines 137-139 to clarify this. 

8) why 6-week old mouse, instead of 14 or 21d old mouse? For instance, wouldn't alfalfa benefit the early life gut colonization?

Author Response: Previous research using C. rodentium as a challenge pathogen indicates that 18-20g mice are most susceptible to infection. Female mice at 6 weeks of age were selected because they were likely to be at this weight after the 2-week diet adaptation period. This is in accordance with the Nature protocol by Crepin et al. 2016 (manuscript citation 38) which is the basis for the experimental work for the Citrobacter challenge. This information is added to new lines 111-113. 

9) Flow cytometry analysis did not include the aqueous extract treatment? Why?

Author Response: Due to the large amount of data generated in this study, discussion of immune cell populations and the microbiota were limited to treatments where changes to these systems could be connected to the BW and ADFI phenotypes. As such, neither the hay nor aqueous extract treatments met these criteria and were not included in the results/discussion for this manuscript. This reasoning is presented in lines 82-84 and again in lines 552-554.

Results:

10) Please, re word the sentences on lines 253-255, that's confusing

Author response: Sentence split into 2 concise statements for clarity on new lines 366-367

11) the examining recovery method was not explained in material and methods, but it was discussed. Please, add a paragraph explaining the recovery examination post-infection.

Author response: Sentence explaining recovery examination and justification added to new lines 130-132.

12) "Production of TNF-� is often associated with macrophage activity", please delete. TNFa is associated with a variety of activities.

Author response: Mention of macrophage-specific activity is relevant to the cell population being examined. Added “among other outcomes” to new line 351-352 to convey that activity of this cytokine is not limited to macrophages. 

13) Please, correct "(PCoA) " to PCA

Author Response: This abbreviation occurs only once in the manuscript and was removed. 

Discussion:

14) Please, reduce the discussion about BW. It's too long for 2 parameters.

Author response: Discussion of these parameters shortened and compiled into one paragraph on new lines 536-554

---

## [Decision Letter · Decision Letter 1]

30 Jun 2020

Host immunity and the colon microbiota of mice infected with Citrobacter rodentium are beneficially modulated by lipid-soluble extract from late-cutting alfalfa in the early stages of infection

PONE-D-20-06132R1

Dear Dr. Bobeck,

We’re pleased to inform you that your manuscript has been judged scientifically suitable for publication and will be formally accepted for publication once it meets all outstanding technical requirements.

Kind regards,

Juan J Loor

Academic Editor

PLOS ONE

Additional Editor Comments (optional):

Reviewers' comments:

Reviewer's Responses to Questions

**Comments to the Author**

1. If the authors have adequately addressed your comments raised in a previous round of review and you feel that this manuscript is now acceptable for publication, you may indicate that here to bypass the “Comments to the Author” section, enter your conflict of interest statement in the “Confidential to Editor” section, and submit your "Accept" recommendation.

Reviewer #1: All comments have been addressed

2. Is the manuscript technically sound, and do the data support the conclusions?

Reviewer #1: Yes

3. Has the statistical analysis been performed appropriately and rigorously? 

Reviewer #1: Yes

4. Have the authors made all data underlying the findings in their manuscript fully available?

Reviewer #1: Yes

5. Is the manuscript presented in an intelligible fashion and written in standard English?

Reviewer #1: Yes

6. Review Comments to the Author

Reviewer #1: (No Response)

7. PLOS authors have the option to publish the peer review history of their article (what does this mean?). If published, this will include your full peer review and any attached files.

Reviewer #1: No

---

## [Editor Report · Acceptance letter]

6 Jul 2020

PONE-D-20-06132R1 

Host immunity and the colon microbiota of mice infected with Citrobacter rodentium are beneficially modulated by lipid-soluble extract from late-cutting alfalfa in the early stages of infection 

Dear Dr. Bobeck:

I'm pleased to inform you that your manuscript has been deemed suitable for publication in PLOS ONE. Congratulations! Your manuscript is now with our production department. 

Kind regards, 

on behalf of

Dr. Juan J Loor 

Academic Editor

PLOS ONE